# Learning from Inside: Self-driven Siamese Sampling and Reasoning for Video Question Answering

**Weijiang Yu**[1], **Haoteng Zheng**[1], **Mengfei Li**[1], **Lei Ji**[2], **Lijun Wu**[2], **Nong Xiao**[1]*, **Nan Duan**[2]

[1]School of Computer Science and Engineering, Sun Yat-sen University
[2]Microsoft Reasearch Asia
weijiangyu8@gmail.com, xiaon6@mail.sysu.edu.cn
{zhenght5, limf33}@mail2.sysu.edu.cn
{leiji, lijun.wu, nanduan}@microsoft.com

## Abstract

Recent advances in the video question answering (i.e., VideoQA) task have achieved strong success by following the paradigm of fine-tuning each clip-text pair *independently* on the pretrained transformer-based model via supervised learning. Intuitively, multiple samples (i.e., clips) should be *interdependent* to capture similar visual and key semantic information in the same video. To consider the interdependent knowledge between contextual clips into the network inference, we propose a **Sia**mese **Sam**pling and **Rea**soning (**SiaSamRea**) approach, which consists of a siamese sampling mechanism to generate sparse and similar clips (i.e., siamese clips) from the same video, and a novel reasoning strategy for integrating the interdependent knowledge between contextual clips into the network. The reasoning strategy contains two modules: (1) siamese knowledge generation to learn the inter-relationship among clips; (2) siamese knowledge reasoning to produce the refined soft label by propagating the weights of inter-relationship to the predicted candidates of all clips. Finally, our SiaSamRea can endow the current multimodal reasoning paradigm with the ability of learning from inside via the guidance of soft labels. Extensive experiments demonstrate our SiaSamRea achieves state-of-the-art performance on five VideoQA benchmarks, e.g., a significant **+2.1%** gain on MSRVTT-QA, **+2.9%** on MSVD-QA, **+1.0%** on ActivityNet-QA, **+1.8%** on How2QA and **+4.3%** (action) on TGIF-QA.

## 1 Introduction

By inferring the correct answers for video-based questions, video question answering (VideoQA) has attracted increasing research attention due to its huge application potential, as a fundamental technique for vision-to-language reasoning. The task involves acquisition and manipulation of spatio-temporal visual representations guided by the compositional semantics of the linguistic clues [32, 15, 21, 34]. Existing works can roughly be divided into two aspects. One aspect is to explore a powerful multimodal transformer-based network [22, 2, 34, 45] trained on large-scale datasets (e.g., COCO Captions [3] and HowTo100M [29]).

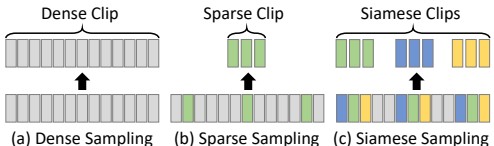

Figure 1: Different sampling mechanisms for video frames. (a) Traditional methods use dense clip features from full-length videos. (b) A recent approach [22] suggests sparsely sampled clips for end-to-end learning. (c) Our siamese sampling to generate similar semantic clips .

---

*Corresponding Author: xiaon6@mail.sysu.edu.cn

35th Conference on Neural Information Processing Systems (NeurIPS 2021).

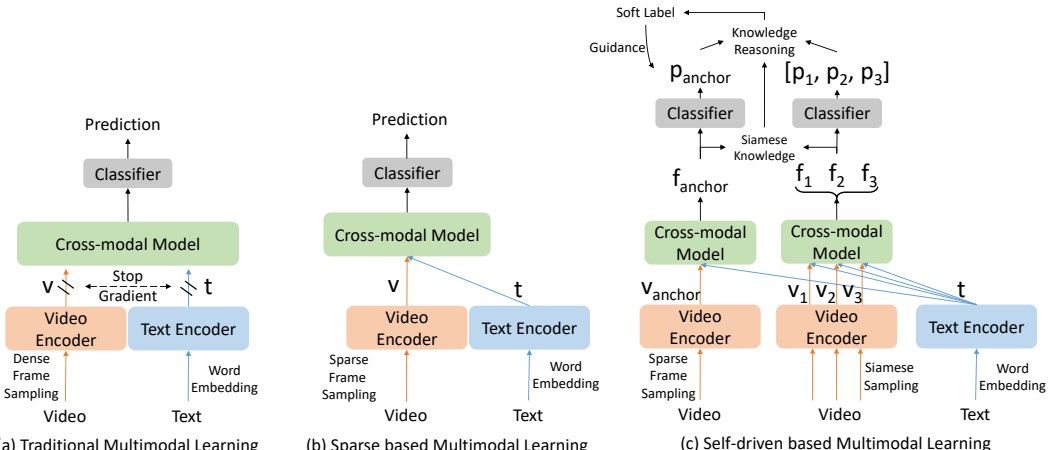

Figure 2: (a) Traditional multimodal learning uses dense sampling for video frames and extracts individual clip-text features via offline encoders. (b) Sparse based multimodal learning benefits from sparsely sampled clips (independent) and raw text tokens for end-to-end modeling. (c) Self-driven based multimodal learning utilizes cross-relationship between anchor clip ($\mathbf{v}_{anchor}$) and siamese clips ($\mathbf{v}_1, \mathbf{v}_2, \mathbf{v}_3$) as knowledge to lead the model to create soft labels for self-supervised learning. Note that the modules in (c) using same color mean the weights are shared.

The other aspect aims at exploring the structure reasoning for semantic alignment between vision and language (e.g., Hierarchical Reasoning [21], Heterogeneous Graph Alignment [15, 46]and Object Relation Reasoning [16]). Both of them solely consider each clip-text pair separately and ignore the correlation between contextual clips in the same video.

By further analyzing the existing multimodal learning paradigm and showing their difference in Figure 2, we observe that current methods suffer from a key drawback: every clip-text pair is regarded as *individual* and *independent* during the training. Such drawback overlooks the rich interaction of contextual clips from the same video[2]. We believe that the internal interaction information from same videos can be helpful for further enhancing the network learning. Hence, to provide a remedy to this dilemma, we present a self-driven Siamese Sampling and Reasoning (SiaSamRea) framework *learning from inside*, by using the internal contextual semantics of *interdependent* video-aware data (e.g., clips) from the same video in the training process.

First of all, our SiaSamRea (in Figure 3) consists of two key parts: (**1**) a siamese sampling shown in Figure 1 (c) to extract multiple similar clips from the same video, which is motivated by ClipBert [22]; (**2**) a reasoning strategy named self-driven based multimodal learning as shown in Figure 2 (c). The strategy contains two modules: ($i$) a siamese knowledge generation module to calculate the correlation matrix between the anchor clip-text pair (e.g., $\mathbf{f}_{anchor}$) and siamese clip-text pairs (e.g., $\mathbf{f}_1, \mathbf{f}_2, \mathbf{f}_3$); ($ii$) a siamese knowledge reasoning module to produce soft labels for self-supervised refinement during the training. Finally, the labels are applied as auxiliary training supervision to enhance the network.

Compared with previous sampling mechanism, our siamese sampling as shown in Figure 1 (c) not only is sparse but also is able to generate multiple similar clips for constructing their internal relationships. Specifically, our siamese sampling captures clips at different start frames with same interval time in the same video, which can constrain the global semantic of each clip to be similar (i.e., each clip can represent the consistent video content from a global perspective).

Furthermore, to fully utilize the siamese clips, we explore a new reasoning strategy namely self-driven based multimodal learning as shown in Figure 2 (c). There are three steps in the strategy for using the interaction of internal clips from the same video into the training. At the $1^{st}$ step, the anchor clip and siamese clips are obtained by using sparse sampling and siamese sampling, respectively. Then the two types of clips individually cooperated with the text are fed into the model to extract clip-text features, including anchor clip-text feature (e.g., $\mathbf{v}_{anchor}$) and siamese clip-text features (e.g., $\mathbf{v}_1, \mathbf{v}_2, \mathbf{v}_3$). At the $2^{nd}$ step, the internal contextual interaction[3] between the anchor clip-text feature and siamese

---

[2]We call the contextual clips from the same video as *internal clips* containing anchor and siamese clips

[3]The internal contextual interaction is regarded as siamese knowledge in our paper.

clip-text features is calculated via our siamese knowledge generation module. At the $3^{rd}$ step, a siamese knowledge reasoning module is proposed to use the siamese knowledge applying to several predicted candidates (e.g., $\mathbf{p}_{anchor}, \mathbf{p}_1, \mathbf{p}_2, \mathbf{p}_3$) for adaptively reasoning out the refined soft label. Here, the siamese knowledge is applied for adaptively reasoning. Because it is hard to distinguish which candidate is critical to the accuracy from several predictions during the training. Finally, we use the refined soft label to further distill our model for high-quality representation generation.

Our contributions are three-fold: ($i$) We propose a novel end-to-end framework named SiaSamRea for learning from inside on VideoQA task, by using **sia**mese **sam**pling and **rea**soning to integrate the interdependent semantics of clips from the same video into the training process. ($ii$) A novel reasoning strategy is carefully designed for building the soft guidance from the interdependent knowledge between internal clips, which consists of a siamese knowledge generation module and a siamese knowledge reasoning module. ($iii$) Experiments on five commonly-used VideoQA benchmarks show the superior ability of our SiaSamRea and demonstrate the effectiveness of our proposed components. Not that our method only teaches the network with interdependent knowledge during the training, which does not bring any extra burden (e.g., computation, memory and parameters) in the inference.

## 2    Related Work

**Visual Content Modeling on VideoQA.** Video Question Answering is a task aiming to answer the given question concerning video content. Some current works extract generic visual appearance and motion features to represent video contents and design different attention mechanisms to integrate these features, like question-routed attention [38, 14] and co-attention [9, 49]. These methods mainly focus on the holistic understanding of video contents, which easily neglect the meaningful details of local clips. Some research pays attention to structure reasoning for semantic alignment [15, 16, 11, 17, 21]. Jiang et al. [15] proposed to build the correlation between the inter- and intra-modalities for cross-modal learning via a heterogeneous graph alignment framework. Huang et al. [11] presented a novel location-aware graph convolution network to mine the structural representation between the location and relation among visual objects. Le et al. [21] demonstrated the effectiveness of hierarchical video features for VideoQA by developing a general-purpose neural reasoning unit. These works aim to handle fine-grained visual entity (e.g., object) and multimodal hierarchy but ignore the correlation between contextual clips in the same video.

**Pretrained-finetuning Paradigm on VideoQA.** Recently, the paradigm of pretrained to fine-tuning has made significant progress in many multimodal tasks, such as text-video retrieval [33], visual question answering [1], image captioning [44], video captioning [39] and video question answering [23]. Thanks to the strong success of transformer-based [37] language pretraining [6, 27, 41] and image-text pretraining [36, 4, 5, 12, 31], the video-text pretraining [24, 29, 51, 35] has shown promising improvement on the video-language tasks, especially on VideoQA task. For example, Zhang et al. [50] proposed to enhance the visual representation based on the OSCAR [26] by incorporating more visual data during the training. Lei et al. [22] solved the offline encoder in the pretrained-finetuning pipeline by adopting a sparse training and dense inference. To promote the development of large-scale multimodal learning, Seo et al. [34] proposed a new dataset and task for future utterance prediction based on the video-text inputs. The multimodal model trained on their new dataset also performed well on the VideoQA task. It can further prove the transfer ability of the pretrained-finetuning paradigm. All approaches as mentioned-above on VideoQA are benefited from the powerful pretrained backbone with large-scale clip-text pairs, while each clip-text pair is independently encoded into the network rather than mining their contextual knowledge in the same video. In this paper, we argue that the rich contextual information among clips should be valuable and can well enhance the network. Hence, we propose a siamese sampling and reasoning method for learning from internal clips in the same video as well as bringing no additional burden in the inference, which consists of a siamese sampling mechanism and a reasoning strategy.

## 3    Methodology

Our Siamese Sampling and Reasoning (SiaSamRea) method is shown in Figure 3. The SiaSamRea can be roughly divided into three parts based on the learning pipeline: (1) Clip Sampling; (2) Feature Extraction; (3) Reasoning Strategy. The clip sampling aims to get the anchor clip and siamese clips by sparse sampling and siamese sampling, respectively. The feature extraction is composed of a video

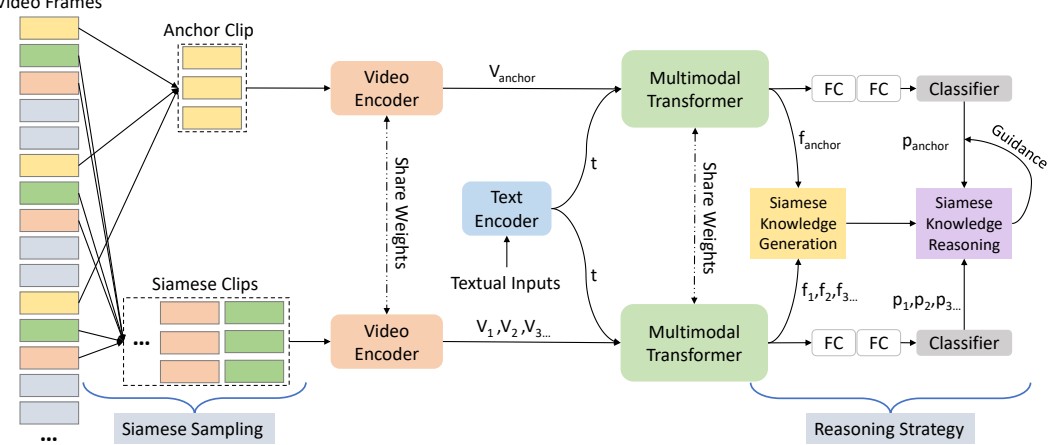

Figure 3: Overview of our Siamese Sampling and Reasoning (SiaSamRea) applied to VideoQA. Given the dense video frames, the anchor clip and siamese clips first to be extracted by sparse sampling and siamese sampling. Then a video/text encoder and a multimodal transformer are utilized to generate multimodal features ($\mathbf{f}_{anchor}, \mathbf{f}_1, \mathbf{f}_2, \mathbf{f}_3$). Next, a siamese knowledge generation module is proposed to produce contextual relationship between anchor clip and siamese clips from the same video. Finally, the soft label used for distilling the whole network, can be obtained via siamese knowledge reasoning module by inferring the siamese knowledge on several candidates ($\mathbf{p}_{anchor}, \mathbf{p}_1, \mathbf{p}_2, \mathbf{p}_3$). Note that we only need the network fed with anchor clip for inference.

encoder, a text encoder and a multimodal transformer, which encodes the multiple clip-text pairs as semantic representations. The goal of the reasoning strategy is to generate refined labels for distilling the model learning, which can be achieved by using the siamese knowledge reasoning module to propagate the siamese knowledge (i.e., interdependent relationship among pairs) from the siamese knowledge generation module into the predictions of multiple clip-text pairs. Our contribution mainly focuses on the design of reasoning strategy in a single framework, the siamese sampling mechanism and innovative modules (e.g., siamese knowledge generation and siamese knowledge reasoning), which are unveiled and discussed in details in the following sub-sections.

## 3.1 Preliminary

To simplify our method, we unify some symbolic notations in this section. Given the dense video frames $\mathbb{F}$, we seek to construct the anchor clip $\mathbf{c}_{anchor}$ and siamese clip set $\mathcal{C}^{siamese} = \{\mathbf{c}_i\}_{i=1}^{N-1}$, where the $N$ means all clips number. Each clip $\mathbf{c}$ is uniformly sampled $B$ frames and obtains $B$ feature maps by using the video encoder to encode the clip. The encoded features of anchor clip and the siamese clip set are denoted as $\mathbf{v}_{anchor}$ and $\mathcal{V}^{siamese} = \{\mathbf{v}_i\}_{i=1}^{N-1}$, correspondingly. Taking the encoded text representation from the text encoder concatenated with the clip representation from the video encoder as input, a multimodal transformer is applied to produce the clip-text feature. Each anchor clip-text feature is indicated as $\mathbf{f}_{anchor}$ and the associated siamese clip-text feature set indicates $\mathcal{F}^{siamese} = \{\mathbf{f}\}_{i=1}^{N-1}$. Similarly, their predictions are defined as $\mathbf{p}_{anchor}$ and $\mathcal{P}^{siamese} = \{\mathbf{p}_i\}_{i=1}^{N-1}$, respectively. The $\mathbf{p}$ is a vector with $K$ dimension according to the classification number.

## 3.2 Siamese Sampling

In this section, we introduce our siamese sampling as shown in Figure 1 (c), which is inspired by the spare sampling in ClipBert [22]. First, we get an anchor clip $\mathbf{c}_{anchor}$ by sparsely and uniformly sampling $B$ frames from dense video frames $\mathbb{F}$. The length of $\mathbb{F}$ is typically much larger than $B$. We randomly decide the starting sampling index of the anchor clip. Then our siamese sampling captures siamese clips at different starting indexes but nearby the index of the anchor clip in the same video. We use the same interval frame for all clips. There are some constraints for selecting the starting sampling index of the anchor clip. It should be in the first third of the video to guarantee the equilibrium of sampling. The "nearby" means the index of the Siamese clip should be next to the index of the anchor clip. The pace of sampling is consecutive before or after the anchor index. After siamese sampling, we can obtain some siamese clips, which have similar global video semantics

including the anchor clip. Then we feed these clips into the video encoder sequentially as shown in Figure 3 to get the visual feature. The video encoder and text encoder are similar with the encoders in the ClipBert [22].

## 3.3 Reasoning Strategy

Different from previous methods [42, 22, 51, 43] that directly utilized the clip-text features $\mathbf{f}$ from multimodal transformer to predict the final results, we propose a novel reasoning strategy named self-driven multimodal learning by exploring how to integrate knowledge between different clips (i.e., siamese clips) in the same video with a single network, which consists of a siamese knowledge generation module and a siamese knowledge reasoning module.

### 3.3.1 Siamese Knowledge Generation

Intuitively, clips with high visual similarities are expected to have more consistent predictions on their predicted class probabilities, regardless of their ground-truth labels. In our solution, similar clips' knowledge is systematically aggregated and integrated to provide better soft labels.

We propose to propagate and integrate knowledge among clip-text pairs on-the-fly in terms of their multimodal feature similarities. Given a set of $N$ clip-text pairs[4] from the same video and a feature extractor $F$ under training, we first estimate the samples' pairwise similarities by the dot product of their encoded representations with the current network. Such similarities can be stored in an interdependent matrix $\mathbf{A} \in \mathbb{R}^{N \times N}$ as

$$\mathbf{A}(i,j) = \sigma(F(\mathbf{f}_i))^\top \sigma(F(\mathbf{f}_j)), \tag{1}$$

where $i, j$ are the indices of samples in a video and $\sigma$ denotes the $\ell_2$-norm function. $N$ is the number of samples. To avoid reasoning in the self-loop reinforcement, we discard the diagonal entries from $\mathbf{A}$ by $\mathbf{A} = \mathbf{A} \odot (1 - \mathbf{I})$, where $\mathbf{I}$ is an identity matrix and $\odot$ denotes the Hadamard product. Subsequently, we normalize each row of the interdependent matrix $\mathbf{A}$ so that $\sum_{j=1}^{N} \hat{\mathbf{A}}(i,j) = 1$ for all $i$, while keeping the diagonal all zeros, i.e., $\hat{\mathbf{A}}(i,i) = 0$. The normalization can be formulated as a softmax function over each row of the matrix $\mathbf{A}$

$$\hat{\mathbf{A}}(i,j) = \frac{\exp(\mathbf{A}(i,j))}{\sum_{j \neq i} \exp(\mathbf{A}(i,j))}, \quad \forall i \in \{1, \ldots, N\}, \tag{2}$$

where the $\hat{\mathbf{A}}(i,j)$ indicates the normalized interdependent knowledge between the $i$-th sample and $j$-th sample in the same video. The whole interdependent knowledge of all sampling clips denoted as $\hat{\mathbf{A}}$ is also called as siamese knowledge in this paper. Due to the knowledge is mainly calculated by our siamese samples.

### 3.3.2 Siamese Knowledge Reasoning

We denote the predicted probabilities of samples within a video as $\mathbf{P} = [\mathbf{p}_1, \ldots, \mathbf{p}_N]^\top \in \mathbb{R}^{N \times K}$, which satisfy $\sum_{k=1}^{K} P(i,k) = 1, \forall i$. For the $i$-th sample in the video, there are $K$-class candidates. We would like to adaptively propagate and merge the other samples' predictions to create a better soft label for it based on the inter-sample affinities, which can be formulated as

$$\hat{\mathbf{p}}_i = \sum_{j \neq i} \hat{\mathbf{A}}(i,j)\mathbf{p} = \hat{\mathbf{A}}(i)\mathbf{P}, \tag{3}$$

where $\hat{\mathbf{p}}_i$ is the propagated probability vector for the $i$-th sample and can serve as the refined soft labels. Intuitively, if the $i$-th sample and the $j$-th sample are similar with a high interdependent value $\hat{\mathbf{A}}(i,j)$, the prediction $\mathbf{p}_j$ would have a larger weight to be propagated to $\hat{\mathbf{p}}_i$. Motivated by the graph convolutional network [18], our module adaptively propagate the predictions between all the samples (i.e., clip set) in a video in parallel, which can be formulated as

$$\hat{\mathbf{P}} = \mathbf{W}\hat{\mathbf{A}}\mathbf{P}, \tag{4}$$

where the $\mathbf{W} \in \mathbb{R}^{1 \times N}$ is learnable matrix and $\hat{\mathbf{P}} \in \mathbb{R}^K$ is our soft label. To avoid propagating and integrating noisy and unexpected predictions too much, we produce the soft learning targets $\mathbf{Q}$ as a

---

[4]To simplify the expression, the "clip-text pair" denotes "sample" in Sec. 3.3

weighted sum of the initial probability matrix $\mathbf{P}$ and the propagated one $\hat{\mathbf{A}}\mathbf{P}$,

$$\mathbf{Q} = \omega\mathbf{W}_1\hat{\mathbf{A}}\mathbf{P} + (1 - \omega)\mathbf{W}_2\mathbf{P}, \tag{5}$$

where $\omega \in [0, 1]$ is the weighting factor and $\sum_{k=1}^{K} \mathbf{Q}(k) = 1$, $\mathbf{Q} \in \mathbb{R}^K$. The $\mathbf{W}_1$ and $\mathbf{W}_2$ are trainable weights. With the above formulations, the knowledge between samples within the same video can be propagated to each other and integrated for one iteration.

### 3.4  Optimization

**Open-ended VideoQA Setting.** Open-ended question setting is to choose one correct answer from a predefined answer set $\Omega$, which can be seen as a multi-label classification task and trained with the cross-entropy loss function. We feed the visual representation $\mathbf{v}$ and question representation $\mathbf{q}$ into our framework denoted as $\phi_\theta$ to compute prediction probabilities:

$$\mathbf{p} = \phi_\theta(\mathbf{v}, \mathbf{q}), \mathbf{p} \in \mathbb{R}^{|\Omega|}. \tag{6}$$

**Multiple-choice VideoQA Setting.** Multiple-choice question setting is to choose one correct answer from $M$ candidates. In this case, we first formulate the answer representation of the $m$-th candidate as $\mathbf{a}_m$. Then, we feed the visual representation, question representation and answer representation into our model to ouput the $m$-th answer score, which is formulated as

$$s_m = \phi_\theta(\mathbf{v}, \mathbf{q}, \mathbf{a}_m), 1 \leq m \leq M, \tag{7}$$

where the score of the correct candidate is the positive score $s^p$, and the rest scores are negative scores denoted as $(s_1^n, ..., s_{M-1}^n)$. During training, we utilize the summed pairwise hinge loss $\sum_{i=1}^{M-1} max(0, 1 - (s^p - s_i^n))$ between the positive score and each negative score to train our model.

**Objective Function.** There are two terms for our final objective function $\mathcal{L}$. One term denoted as $\mathcal{L}_{siamese}$ comes from the loss between our soft label and the prediction result $\mathbf{p}_{anchor}$ from the anchor clip-text pair. We utilize the cross-entropy loss function to compute the loss $\mathcal{L}_{siamese}$. The other term is to optimize the training process between predictions and the ground-truth, which is commonly-used in the previous works. Note that the last term may be different depended on the task setting, we uniformly formulate the loss function associated with the ground-truth as $\mathcal{L}_{gt}$. Finally, our model can be finally optimized by

$$\mathcal{L} = \alpha\mathcal{L}_{siamese} + \mathcal{L}_{gt}, \tag{8}$$

where $\alpha$ is a hyper-parameter term to adjust the balance of the two losses.

## 4  Experiment

In this section, we first introduce the data that we use in Sec. 4.1 and details of our experiments in Sec. 4.2. Then we validate the effectiveness of our proposed components in Sec. 4.3, which is followed by the comparison with other methods in Sec. 4.4. Finally, in Sec. 4.5, some visualization results are shown to qualitatively analyze the benefits of our siamese clips.

### 4.1  Evaluation and Datasets

We evaluate our proposed VideoQA method and compare it with other state-of-the-art methods on five VideoQA datasets, which are widely used and accepted for academic video question answering, including the open-ended setting and multiple-choice setting. We follow the previous evaluation protocols for open-ended settings [21, 48] and utilize a fixed vocabulary of training answers. For the multiple-choice evaluation setting, we follow the previous methods [21] to use the same dataset split setting. Excepting for the repetition count task in the TGIF-QA [13] where the Mean Squre Error (MSE) is excavated, we use top-1 accuracy to be the evaluation metric for all experiments.

We carry out our SiaSamRea on three open-ended VideoQA datasets, including MSRVTT-QA [38], MSVD-QA [38] and ActivityNet-QA [48]. The MSRVTT-QA has 10K videos and 244K question-answer pairs. The size of the predefined answer set is 1000. Compared with the MSRVTT-QA, the MSVD-QA is smaller, which has 2K videos with nearly 51K question-answer pairs. The MSRVTT-QA and MSVD-QA contain five question types like what, who, how, when and where. ActivityNet-QA has 5.8K videos and 58K questions, which is repurposed from ActivityNet Captions [19].

Furthermore, we validate our method on a multiple-choice VideoQA datasets, such as How2QA [24]. The How2QA has 9K videos tailed with 44K question numbers, where each question is associated with one correct and three incorrect answers. Besides, we train and test our model on the TGIF-QA [13], which is a large-scale VideoQA dataset containing 72K animated GIFs and 165K question-answer pairs. There are four tasks defined on this dataset: ($i$) Repetition Count is an open-ended QA task to count the number of the repetitions of an action; ($ii$) Repeating Action is similar to the repetition count task but following the multiple-choice setting; ($iii$) State Transition is multiple choice task for identifying the transition between two states; ($iv$) Frame QA is open-ended to find the sufficient frame to answer the questions.

## 4.2 Implementation Details

We implement our proposed SiaSamRea via PyTorch [30], a well-known open-source deep learning framework. For being fairly comparable with other methods, we conduct the following training and testing settings. Unless otherwise noted, settings are the same for all experiments.

**Training.** We use the pretrained model trained on the COCO Captions [3] and Visual Genome Captions [20] from ClipBert [22] as our initial weights of our SiaSamRea. To obtain more VideoQA-related pretrained weights, we present to train task-related data during the pretraining. Based on the initial weights from the ClipBert, we perform the clip-text pretraining on TGIF-QA [13] optimized by adaptively using aforementioned objection function (i.e., Eq. 8) according to different task settings (i.e., open-ended and multiple-choice tasks). There is a total of 71739 videos in the TGIF-QA dataset. Because our proposed method aims to model the long-range contextualized knowledge via Siamese samples. The videos with less than 40 frames do not meet our requirements in the training. We filter out less than 40 frames of video during the pre-training to remain long-range temporal videos. In the TGIF-QA dataset, the percentage of videos with less than 40 frames is 56% (40333/71739). Hence, we use the remaining 31406 TGIF videos for our pre-training, which is less than the ClipBERT. The ClipBERT uses a total of 71739 TGIF-QA videos to finetune the four tasks, which is different from our pre-training on the TGIF-QA dataset. Because the video length is various, we set the number of frames in each clip from 5 to 16 according to the video length. We set the batch size to be 1 per GPU and use AdamW [28] optimizer with initial learning rate 0.0005. The learning rate warm-up strategy is adopted over the first 10% training steps followed by linear decay to zero. We terminate the pretraining when it reaches minimal learning rate (i.e., 0.00001) and its validation performance stabilizes (i.e., reaches convergence). The balance term is $\alpha$=1.0, the integrating weight is $\omega$=0.5. We conduct all experiments using 16 NVIDIA V100 GPU cards.

For the downstream fine-tuning, we apply the same training and optimizer configurations as mentioned-above. Since the downstream datasets are various and in multiple domains, we use dataset-specific learning rates, training epochs and clip numbers based on validation performance. Note that we do not need to excavate additional fine-tuning on the TGIF-QA dataset.

**Testing.** We strictly follow the test split setup in each dataset and validate the learning outcomes after each learning epoch. Then we report the best metrics (top-1 accuracy) as our results.

## 4.3 Ablation Studies

We conduct comprehensive ablation studies concerning various aspects of our SiaSamRea's design in this section. In Table 1, we set the network training without using the $\mathcal{L}_{siamese}$ as our baseline. Compared with the baseline, our method only using the siamese knowledge reasoning module (w/ SKR) by replacing the siamese knowledge generation with average operation, can apparently boost the accuracy by more than 4% from baseline on How2QA and

Table 1: Ablation studies on How2QA and MSVD-QA datasets. The SKG and SKR separately indicate siamese knowledge generation and siamese knowledge reasoning.

| Methods | How2QA | MSVD-QA |
|---|---|---|
| baseline | 79.1 | 39.4 |
| w/ SKR | 83.0 | 44.7 |
| w/ SKG + SKR | **84.1** | **45.5** |

MSVD-QA. It can demonstrate the effectiveness of our siamese knowledge reasoning. It also validates our assumption that multiple clips from same videos can really bring some strong knowledge to enhance the network ability. When we add the siamese knowledge generation module (w/ SKG+SKR), our method can reach a higher performance to 84.1% and 45.5%, which can demonstrate the effectiveness of building the interdependent knowledge (i.e., siamese knowledge) for integrating the samples. It can also prove the adaptively reasoning by our siamese knowledge is better than the purely

average operation. We think that the siamese knowledge not only serves as the knowledge-routed representation, but also implicitly constrains the semantic consistency of clips in the space of clip-text features.

*To explore the effect of the number of siamese clips*, we compare our method with various number of siamese clips, which results can be shown in Table 2. Note that our siamese samples are serving for the interdependent knowledge extraction. Intuitively, more samples can aggregate richer knowledge. When adding the number of siamese samples from 1 to 12, our method promotes the accuracy from 79.6% to 84.4% on How2QA and 39.9%→45.7% on MSVD-QA. Such improvement can demonstrate our intuition that more siamese samples can bring richer knowledge, which makes our network benefited from it. Although the accuracy is increasing with the number of siamese clips, we observe that the improvement from the number 8 to 12 is slight. We think the reason is the saturation of knowledge, i.e., the model has enough knowledge to learn the task on the two datasets. Hence, it is almost meaningless to purely increase the siamese samples. To balance the training time and accuracy, we select the 8 siamese samples in our final version.

Table 2: The effect of the number of sampling siamese clips on How2QA and MSVD-QA.

| Methods | 1 | 2 | 4 | 8 | 12 |
|---|---|---|---|---|---|
| How2QA | 79.6 | 80.5 | 81.9 | 84.1 | 84.4 |
| MSVD-QA | 39.9 | 41.3 | 42.7 | 45.5 | 45.7 |

*The effect for W1 and W2 in Eq. 5.* We have conducted experiments on the MSVD-QA dataset to analyze the effect of these learnable weights (W1 and W2). Our SiaSamRea without W1 performs 44.9% on MSVD-QA, which heavily drops 0.6% performance compared with our whole SiaSamRea (45.5%). One possible reason is that the W1 can make the propagated prediction matrix learnable and adaptively propagate the predictions between all the samples. When performing our model without W2, it achieves 45.2%. The performance without W2 is slightly decreasing ( 0.3%). When it comes to our method without using W1 and W2, it gets 44.5%. As we can see, the W1 and W2 both can bright improvement. Hence, it can further demonstrate the importance and necessity for the generation of expected soft labels by correctly merging several predicted candidates.

Table 3: Comparison with state of the art on MSRVTT-QA and MSVD-QA (top-1 accuracy).

| Methods | MSRVTT-QA | MSVD-QA |
|---|---|---|
| E-SA [38] | 29.3 | 27.6 |
| ST-TP [13] | 30.9 | 31.3 |
| AMU [38] | 32.5 | 32.0 |
| Co-mem [9] | 32.0 | 31.7 |
| HME [7] | 33.0 | 33.7 |
| LAGCN [11] | — | 34.3 |
| HGA [15] | 35.5 | 34.7 |
| QueST [14] | 34.6 | 36.1 |
| MiNOR [16] | 35.4 | 35.0 |
| TSN [40] | 35.4 | 36.7 |
| HCRN [21] | 35.6 | 36.1 |
| Clip-BERT [22] | 37.4 | — |
| SSML [2] | 35.1 | 35.1 |
| CoMVT [34] | 39.5 | 42.6 |
| SiaSamRea (Ours) | **41.6** | **45.5** |

*The effect for multiple copies of anchor clip.* We replace our Siamese clips with multiple copies of anchor clips to analyze the effect of SKG. Since we used 8 siamese clips in our paper, we replaced the siamese clips with 8 copies of anchor clips during the training. We execute this ablation study on the MSVD-QA dataset. This ablation study achieves 41.3%. It promotes the baseline (39.4%) with 1.9% performance. It can validate the power of the ensemble way. But only using the anchor clips for ensemble can't bring visually contextualized knowledge. We skillfully exploit the visually contextualized knowledge via the siamese clips, which obtains 45.5% accuracy. Hence, the improvement of our method mainly comes from the SKR and SKG by using Siamese clips rather than the simple ensemble.

### 4.4 Comparisons with State-of-the-Arts

**Comparisons on MSRVTT-QA and MSVD-QA.** Table 3 reports the comparison with previous methods on MSRVTT-QA and MSVD-QA datasets. Our SiaSamRea achieves the best performance on both datasets. Specifically, compared with the methods using large-scale pretrained data (i.e., Clip-BERT, SSML and CoMVT), our method consistently and considerably outperforms them by 2.1% and 2.9% top-1 accuracy on MSRVTT-QA and MSVD-QA when compared with CoMVT, which was previously the best performer. When it comes to other methods (e.g., HCRN, TSN etc.) without using additional pretrained data, our model can significantly boost the accuracy by around 6% accuracy on MSRVTT-QA and a large gain of 9.4% on MSVD-QA compared with HCRN. These promising results can validate the effectiveness of using pretrained data. It also supports the feasibility of distilling the interdependence of multiple clips with our reasoning strategy on the open-ended VideoQA task.

**Comparisons on ActivityNet-QA and How2QA.** We now move to the evaluation on ActivityNet-QA and How2QA, which results are reported in Table 4. In particular, our method improves over the recent CoMVT approach that has been pretrained on HowTo100M [29] dataset. It can effectively demonstrate the advantages of applying related-data (e.g., TGIF-QA) on the pretraining, which can perform well to transfer to related tasks. The ActivityNet-QA is so challenging due to the complex events and backgrounds in the videos. Our SiaSamRea still achieves a new state-of-the-art performance (39.8%) on ActivityNet-QA by learning from internal clips. These strong results show the importance and powerful ability of our proposed learning strategy. In other words,

Table 4: Comparison with state of the art on ActivityNet-QA and the public val set of How2QA (top-1 accuracy).

| Methods | ActivityNet-QA | How2QA |
|---|---|---|
| E-SA [48] | 31.8 | — |
| MAR-VQA [52] | 34.6 | — |
| HERO [24] | — | 74.1 |
| CoMVT [34] | 38.8 | 82.3 |
| SiaSamRea (Ours) | **39.8** | **84.1** |

our SiaSamRea can provide another way to handle the complex situation by mining the intrinsic information like internal relationship to parse the various temporal scenes from multiple samples. On the multiple-choice task, the ability of our method gets great scores (84.1%) on How2QA dataset as validating the availability of transferring our method to the other video question answering settings.

Table 5: Comparison with the state-of-the-art methods on TGIF-QA dataset. For count, the lower the better.

| Methods | Action | Trans. | Frame | Count |
|---|---|---|---|---|
| VIS+LSTM (agg) [32] | 46.8 | 56.9 | 34.6 | 5.09 |
| VIS+LSTM (avg) [32] | 48.8 | 34.8 | 35.0 | 4.80 |
| VQA-MCB (agg) [8] | 58.9 | 24.3 | 25.7 | 5.17 |
| VQA-MCB (avg) [8] | 29.1 | 33.0 | 15.5 | 5.54 |
| CT-SAN [47] | 56.1 | 64.0 | 39.6 | 5.13 |
| ST-TP [13] | 62.9 | 69.4 | 49.5 | 4.32 |
| GR-ATT [38] | 68.8 | 73.9 | 53.0 | 4.32 |
| Co-mem [9] | 68.2 | 74.3 | 51.5 | 4.10 |
| PSAC [25] | 70.4 | 76.9 | 55.7 | 4.27 |
| STA [10] | 72.3 | 79.0 | 56.6 | 4.25 |
| MiNOR [16] | 72.7 | 80.9 | 57.1 | 4.17 |
| HME [7] | 73.9 | 77.8 | 53.8 | 4.02 |
| HCRN [21] | 75.0 | 81.4 | 55.9 | 3.82 |
| ClipBERT*[22] | 75.1 | 80.8 | 56.3 | 4.07 |
| HGA [15] | 75.4 | 81.0 | 55.1 | 4.09 |
| SiaSamRea (Ours) | **79.7** | **85.3** | **60.2** | **3.61** |

**Comparisons on TGIF-QA.** To further demonstrate the advantage of SiaSamRea to infer more various and difficult scenarios, we estimate it on the TGIF-QA dataset. The experimental results are summarized in Table 5. It is clear that our SiaSamRea consistently outperforms state-of-the-art models on four reasoning tasks. The four tasks are requiring the strong temporal reasoning especially for the action and transition task. Hence, they really require the model to consider the contextual dependence of clips. Particularly, our method increases overall accuracy on the publicly test set by 4.3% (75.4%→79.7%) compared with the best result on repeating action task, 3.9% (81.4%→85.3%) on state transition task, 3.1% (57.1%→60.2%) on FrameQA task and decreases error by 0.21 (3.82→3.61) on repetition count task. Thanks to the siamese sampling and reasoning, our method can effectively capture the consistent video content from a global perspective. Because our SiaSamRea can distinguish the global semantic of a video by multiple views (i.e., siamese clips), which can bring abundant information from inside that may be absent in other clips to help the network learning the contextual relationship.

To fairly compare the effect of ClipBERT with our proposed method, we conduct extra experiments to finetune the ClipBERT on the filtered data and test it on the four tasks. We mark the ClipBERT finetuned on our filtered data as ClipBERT*. The ClipBERT* gets 75.1% on Action, 80.8% on Trans., 56.3% on Frame and 4.07 on Count. As we can see, the ClipBERT* can achieve comparable results compared with other SOTA methods. It can prove the effectiveness of the pre-trained weights from COCO Captions and Visual Genome Captions. However, our proposed method still outperforms the ClipBERT* on four tasks based on the same filtered data (79.7% on Action, 85.3% on Trans., 60.2% on Frame and 3.61 on Count). It can further demonstrate the superior ability of our idea by using the Siamese clips during the training. Our proposed method achieves better performance with less data. It also validates the huge potential of inside data once there is a suitable way to explore them.

Q: What perched on a branch of tree bit a tail of another animal?    A: Bird

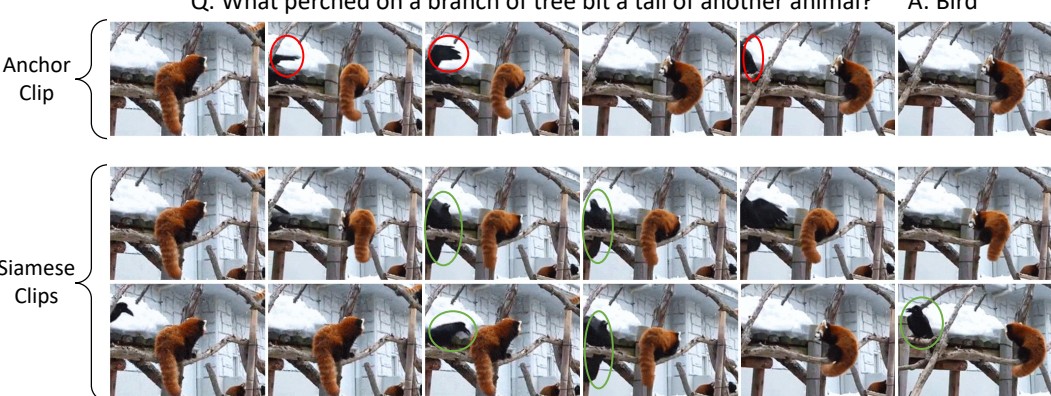

Figure 4: The examples to show the benefits from the siamese clips. It is hard to discriminate the visual content "bird" from ambiguous parts (red circle) in anchor clip. It is much easier to distinguish "bird" by the assistance of siamese clips that contains more complete visual content (green circle).

## 4.5  Visualization

In Figure 4, we show a training example to qualitatively analyze what kind of knowledge does the siamese clips bring to the anchor clip. It is unavoidable to lose some visual contents when sparsely sampling from the video. For example, the visual content "bird" in anchor clip is almost absent. However, we can obtain the absent content from the siamemse clips due to different sampling indices and duration. Hence, our siamese samples can help the model to correctly and stably train on the sparse samples as well as avoiding confusing predictions due to the incomplete visual information.

## 5  Conclusion and Boarder Impact

In this paper, we propose to endow the current multimodal reasoning paradigm with the ability of learning from inside on the VideoQA task via **Sia**mese **Sam**pling and **Rea**soning (SiaSamRea), which contains two key parts: (1) a siamese sampling to produce some sparse clips with similar semantics in the same video; (2) a reasoning strategy to distill the interdependent knowledge between clips into the network. The reasoning strategy is composed of two modules: ($i$) siamese knowledge generation to implicitly aggregate the inter-relationship of clips from the same video; ($ii$) siamese knowledge reasoning to infer soft label by using the predicted candidates of all clips and their inter-relationship. Our proposed SiaSamRea finally can be jointly evolved by the soft label guidance and ground truth, which is evaluated on five VideoQA datasets demonstrating state-of-the-art performance.

This work analyzes an interesting problem of how to *learn from inside* on the video question answering. Different from current methods mainly look for insights from *outside* data and powerful pretrained models. Our work proposes a totally different insight called *learning from inside*. In this paper, we observe the potential power of clips in the same video. Intuitively, the multiple clips should contain rich *interdependent* knowledge which is ignored by current advanced methods. To the best of our knowledge, we are the first one to present learning from inside by using the knowledge between internal clips to assist in evolving the network training. Moreover, our method only teaches the network with interdependent knowledge during the training, which does not bring any extra burden (e.g., computation, memory and parameters) in the inference. Hence, it is a light way to easily equip our method with other suitable networks on different multimodal tasks. We hope that our work will increase interest in the exciting field of learning from visually contextualized knowledge.

## 6  Acknowledgments

We thank all reviewers for providing the thoughtful and constructive suggestions. This work was supported in part by National Natural Science Foundation of China under Grant No.U1811461, in part by Natural Science Foundation of Guangdong Province, China under Grant No.2018B030312002, and in part by the Major Program of Guangdong Basic and Applied Research No.2019B030302002.

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
