# OpenReview forum: "Learning from Inside: Self-driven Siamese Sampling and Reasoning for Video Question Answering"
_NeurIPS.cc/2021/Conference — NeurIPS 2021 Poster_

### Official Review · Reviewer_R2rx · 2021-07-14

**Rating:** 5
**Confidence:** 4

**Summary:**

The paper addresses VideoQA and proposes a new approach: "Siamese Sampling and Reasoning". The method samples multiple sparse clips per video, an anchor clip, and several siamese clips with the same length but different starting frames. Then, pairwise similarities between these clips are computed and used to produce soft learning targets: Intuitively, all clips should produce similar predictions of the answer class. This self-supervised cross-entropy loss is combined with standard cross-entropy between predictions of the anchor clip and the ground truth labels. The method is evaluated on 5 VideoQA benchmarks and shows significant improvements.

**Main Review:**

## Strength

- The method is novel in that it extends ClipBERTs sampling strategy of using several sparse clips with a new loss and new modules over ClipBERTs simple averaging strategy.

- Related work has been analyzed and described thoroughly.

- Strong results in combination with ablations suggest that the proposed method could be helpful for video understanding tasks.


## Weaknesses

### Major points:

1- **Confusing description**:
Authors claim to learn "inter-relationship among clips" (line 12) and state that other methods "consider each clip-text pair separately" (line 36), "each clip-text pair is independently encoded" (line 107). This sounds to me like they would combine several clip-text data points in some way and integrate knowledge between data points. However, the method samples multiple combinations of frames for the same data point, therefore I find these main claims confusing. E.g. ClipBERT simply states in its abstract that "a single or a few sparsely sampled short clips from a video are used at each training step" which I find to be a much more fitting description.

2- **Effect of pre-training**: Authors load pre-trained weights from ClipBERT and then pre-train again on TGIF-QA dataset. They claim that this has advantages (line 335), however, there is no ablation to support this claim: It would be helpful to see the results on MSRVTT-QA and MSVD-QA without pre-training on TGIF-QA dataset.

3- **No comparisons to ClipBERT**:
- ClipBERT results are missing from Table 5 (Performance on TGIF-QA), while they are added in Table 3 (Performance on MSRVTT-QA and MSVD-QA) and no reason is given on why these results are missing. This is important as ClipBERT reports better performance on TGIF-QA than the paper and has a very similar setup (same model, no more than 1 clip during testing, see ClipBERT paper table 8 second-last row.)

- The ablations in Table 1 should be additionally done either on TGIF-QA or on MSRVTT-QA and compared to the ClipBERT result described above.

4- **Hyper-parameters**: Authors use dataset-specific hyperparameters tuning (line 276) which could be another reason for some of the improvements over the ClipBERT method. What was the hyper-parameter tuning approach?

5- **Table 1**: This table reports the ablation study on the effect of SKG and SKR components. However, there is no result with SKG alone. Also, the authors don't report the mean and std of multiple experiments. Would be great if the authors report mean/std and also results with only SKG.


*Given the above concerns, it seems that the main improvement in this paper comes from taking ClipBERT and pretraining it on TGIF-QA before applying it to other datasets. This is contradicting to the story which reasons that the improvements are due to the proposed new modules.*



### Minor points:

1- Motivation is missing as to why the method is evaluated exclusively on VideoQA and not on other tasks like e.g. Video Classification.

2- There are no details on pre-training and fine-tuning time complexity.

3- Architecture details are missing: "multimodal transformer is applied to produce the clip-text feature" (line 133). I would assume the output is a single feature vector obtained from the CLS token, however, this should be specified directly. Otherwise, it would be very hard to reproduce the paper.

**Typos**:
The paper is difficult to follow. The structure of the paper needs to be improved and also there are lots of typos and grammar errors:
- 78 "Not" should be "Note"
- -Figure 3 sentence 1 "to be extracted" sentence is missing a proper verb.
- 129 "all clips number" is not a proper term, should be "number of all clips".
- 129: "Each clip ... is uniformly sampled ... frames" should be rephrased to "frames are sampled per clip" or similar.
- 136 "dimension" should be "dimensions"
- 144 "including" is confusing here. Does this mean: siamese clips have semantics that are similar "to" the anchor clip?
- 153: "predictions on their predicted class probabilities": Duplicate "predict" is incorrect
- 225, 278: "excavate" means physically digging (e.g. with a shovel), would be better to replace it with "use" here.
- 364 "Boarder" should be "Broader"
- 375 "look" should be "looking"

**Time Spent Reviewing:**

4

---

> ### Author Response · Authors · 2021-08-10
> **Answer for minor points**
>
> In this paper, we focus on the video-based multimodal task. The VideoQA is a classical task meanwhile we are familiar with it. The video classification seems to be a good topic. We will carefully consider improving our method on this research topic in the future. We promise to revise typos and grammatical mistakes and polish our paper in the revised version based on your valuable suggestions.

---

> ### Author Response · Authors · 2021-08-10
> **Answer for Table 1**
>
> Because the result with SKG alone is equivalent to the baseline. The SKG should combine with the SKR to support the model learning. When we leave the SKG alone without SKR, the predictions from the Siamese clips can not propagate to the soft label. It is useless to improve our network. In other words, there is meaningless to analyze the SKG alone for our model.

---

> ### Author Response · Authors · 2021-08-10
> **Answer for Hyper-parameters**
>
> We use different numbers of frames for clips according to the dataset. Because there are multiple videos with various lengths. Some datasets mainly contain long-range videos like ActivityNet-QA. Other datasets tend to be short-range videos such as TGIF-QA. To guarantee we can sample a consistent number of clips (anchor clips and Siamese clips), we should adaptively adjust the pace of sampling for different videos based on their length.

---

> ### Author Response · Authors · 2021-08-10
> **Answer for Comparisons to ClipBERT**
>
> There is a total of 71739 videos in the TGIF-QA dataset. Because our proposed method aims to model the long-range contextualized knowledge via Siamese samples. The videos with less than 40 frames do not meet our requirements in the training. We filter out less than 40 frames of video during the pre-training to remain long-range temporal videos. In the TGIF-QA dataset, the percentage of videos with less than 40 frames is ~56% (40333/71739). Hence, we use the remaining 31406 TGIF videos for our pre-training, which is less than the ClipBERT. The ClipBERT uses a total of 71739 TGIF-QA videos to finetune the four tasks, which is different from our pre-training on the TGIF-QA dataset.
>
> To fairly compare the effect of ClipBERT with our proposed method, we conduct extra experiments to finetune the ClipBERT on the filtered data and test it on the four tasks. We mark the ClipBERT finetuned on our filtered data as ClipBERT*. The ClipBERT* gets 75.1% on Action, 80.8% on Trans., 56.3% on Frame and 4.07 on Count. As we can see, the ClipBERT* can achieve comparable results compared with other SOTA methods. It can prove the effectiveness of the pre-trained weights from COCO Captions and Visual Genome Captions. However, our proposed method still outperforms the ClipBERT* on four tasks based on the same filtered data (79.7% on Action, 85.3% on Trans., 60.2% on Frame and 3.61 on Count). It can further demonstrate the superior ability of our idea by using the Siamese clips during the training. Our proposed method achieves better performance with fewer data. It also validates the huge potential of inside data once there is a suitable way to explore them.
>
> We also validate the ClipBERT on other benchmark datasets by using our experimental setting. It achieves 39.5% on MSVD-QA, 38.0% on ActivityNet-QA and 82.5% on How2QA. As we can see, ClipBERT performs comparable results on the VideoQA tasks thanks to their pre-trained knowledge. Our proposed method improves the model by mining the potential information of internal data, which is demonstrated effectively on five datasets.

---

> ### Author Response · Authors · 2021-08-10
> **Answer for effect of pre-training**
>
> We finetune our method on MSRVTT-QA and MSVD-QA without pre-training on the TGIF-QA dataset to analyze the effect of pre-training on VideoQA data. Our proposed method achieves 39.7% on MSRVTT-QA and 43.0 on MSVD-QA. After pre-training on TGIF-QA, our model gets 41.6% and 45.5% w.r.t. MSRVTT-QA and MSVD-QA. Hence, it proves that task-related pretraining benefits the downstream tasks.

---

> ### Author Response · Authors · 2021-08-10
> **Answer for confused descriptions and grammar mistakes**
>
> Thanks for your suggestion, we will carefully consider these descriptions that you mentioned and polish them to be easily read.

---

### Official Review · Reviewer_QHNZ · 2021-07-15

**Rating:** 6
**Confidence:** 3

**Summary:**

In this paper, the authors propose a network architecture to improve the Video QA task. The model learns and aggregates knowledge from siamese (nearby) frames and develop a reasoning strategy for integrating the inter-dependent knowledge between these contextual clips to make the final prediction. Their two modules, first, the siamese knowledge generation first learns inter-relationships among clips (anchor and siamese) and the second module, siamese knowledge reasoning that produces soft labels for self-supervised refinement during the training. They perform extensive experiments on five Video QA datasets to show the ability of their proposed SiaSamRea on both open ended QA and Multiple Choice QA and improve over the state-of-the-art models.


**Limitations And Societal Impact:**

The authors have not discussed any limitations or the social impact of their method or the domain of VideoQA in general. There are some things that could be important to consider while designing models for these datasets and problems. Are the datasets biased towards some specific answers given the questions (for eg, if there’s perching in the question, would the answer always be bird?). This is an important thing to consider if the training and test distributions are different and to reduce bias in learning. Is the model looking at the image to answer these questions or just overfitting to the textual bias? Also, due to pre-training on large scale datasets which are also potentially biased (racial, gender etc.), this might get propagated in the downstream task training and  hence not suitable if such models are to be deployed in real settings. The authors should consider these limitations and discuss them in the paper.


**Main Review:**

The proposed method, Siamese Sampling and Reasoning (SiaSamRea), has three main components. First, the sparse sampling strategy to get sparse frames and the siamese sampling strategy to get the siamese frames. Then, these frames are encoded using video encoders and a multimodal transformer generates multimodal features which are used to produce contextual relationships between the anchor frames and the siamese frames from the same video. Finally, the knowledge reasoning module generates soft labels based on the similarities between frames acquired in the generation module. The model is trained using cross entropy loss or the multi-margin ranking loss based on the task at hand. The proposed method is novel and the paper is well written.

1. Are the weights for the Video Encode /Text Encoder and Multimodal transformer all initialized using the ClipBert model and then fine-tuned on the respective datasets ? It is good to mention what these encoders are eg. resnet 50, 12 layer transformer etc.

2. In section 3.3.1, the matrix A in line 161 is NxN. Is N the total number of anchor and siamese samples in the full video ? f_anchor_1, f_siam_11, f_siam_12, f_siam_13, f_anchor_2, f_siam_21, f_siam_22, f_siam_23 ……. And so on?

3. Did the authors try using the anchor as the center frame and having siamese frames near to both sides of the center frame?
Currently, from the figure and description it seems the siamese frames are samples only at (t_anchor + t_s) locations.

4. During training, as there are more clips/frames for each video sample, how much does it increase the training time and memory usage compared to, for eg.  ClipBERT?

5. Will your method underperform/overperform if attention over frames is done in dense sampling or following methods similar to Long term feature banks [1]?

[1] Wu, Chao-Yuan, et al. "Long-term feature banks for detailed video understanding." Proceedings of the IEEE/CVF Conference on Computer Vision and Pattern Recognition. 2019.


##Post Rebuttal

I have read the authors' response and they address my concerns. I maintain my original rating.

**Time Spent Reviewing:**

3.5

---

> ### Author Response · Authors · 2021-08-10
> **Limitations**
>
> Data biases: Our system tends to distinguish the wedding scene with a white wedding dress. As we know, the white color is a classical topic for western weddings. However, at the eastern wedding, the popular color should be red. And eastern people in the eastern funeral usually wear white clothes. Hence, our system sometimes mistakenly judges the eastern funeral as a wedding.
>
> Although our system has been trained on large-scale datasets, it is unavoidable to import some biases (e.g., eastern and western weddings). It is a huge challenge to solve the data biases. Currently, our system can not handle such a challenge, which is an apparent shortcoming for our method.

---

> ### Author Response · Authors · 2021-08-10
> **Answer for Question 5**
>
> Limited to the rebuttal time and memory usage, we can not validate the attention frames in dense sampling. But we think this operation can further improve the model performance due to more fine-grained information to be learned.

---

> ### Author Response · Authors · 2021-08-10
> **Answer for Question 4**
>
> The training time of our proposed method is nearly 5 times longer than the ClipBERT. And the memory usage during the training is nearly 8 times as much as the ClipBERT. We set the batch size to be 1 during the training.

---

> ### Author Response · Authors · 2021-08-10
> **Answer for Question 3**
>
> There are some constraints for selecting the starting sampling index of the anchor clip. It should be in the first third of the video to guarantee the equilibrium of sampling.  The “nearby” means the index of the Siamese clip should be next to the index of the anchor clip. The pace of sampling is consecutive before or after the anchor index. We will add more details in this section in our final version. Thanks for your reminder.

---

> ### Author Response · Authors · 2021-08-10
> **Answer for Question 2**
>
> The N in section3.3.1 is the total number of anchor and Siamese samples in the full video.

---

> ### Author Response · Authors · 2021-08-10
> **Answer for Question 1**
>
> The video encoder, Text Encoder, and multimodal transformer are all initialized by using the weights of ClipBERT pre-trained on the COCO Captions and Visual Genome Captions. The video encoder is followed the ClipBERT to use the ResNet-50. The transformer is 12-layer.

---

### Official Review · Reviewer_o6HV · 2021-07-16

**Rating:** 5
**Confidence:** 4

**Summary:**

This paper proposes a Siamese Sampling and Reasoning network (SiaSamRea) to model the interdependent semantic across clips in a video. The contribution of the paper is to employ Siamese sampling and calculate the similarities between the multimodal features of clip-text pairs. The Siamese knowledge is then used to generate soft labels which guide the learning process. Experimental results on five Video QA datasets showed the effectiveness of the proposed method against existing works.

**Ethics Review Area:**

["I don’t know"]

**Main Review:**

Strengths:

(1) I like the idea of modeling the interdependent relationship of internal clips. The proposed method is well-motivated and straightforward for implementation.

(2) The proposed method shows improvements across multiple Video QA datasets

Weaknesses:

(1) Although the idea is interesting, the current implementation appears to be an extended version of ClipBERT with Siamese network.

(2) There are several missing details about the model that the authors should explain more clearly:

•	How many anchor clips are used in the proposed model? How does the number of anchor clips affect the performance?
•	In Eq. 5, what is the soft learning target Q used for?

(3) Experiments:
- Why didn’t the authors report the performance of ClipBERT on TGIF-QA dataset in Table 5. As per results from ClipBERT paper, it outperforms the proposed method on most tasks, as result for the Count task is not available. It is important to compare the proposed method with ClipBERT on the TGIF-QA as it is the most prominent dataset for Video QA.

- As SiaSamRea shares some common components with ClipBERT, please explain the performance degrade on the TGIF-QA against ClipBERT.

- The examples in Figure 4 do not show the interdependent relationship between internal clips in the video. Please provide more diverse showcases.

- Ablation studies are not sufficient. Please justify your choices of hyperparameters such as the number anchor clips, the number frames in each clip, the value of the balance term, and the integrating weight.

 - One of the downsides of the Siamese networks is that it is quadratic in time. Please provide more analysis on the memory usage and computation cost of the proposed model.

Minor comment: Page 6, Line 190: multi-label classification -> multi-class classification


**Time Spent Reviewing:**

5 hours

---

> ### Author Response · Authors · 2021-08-10
> **In Eq. 5, what is the soft learning target Q used for?**
>
> The Q in Eq.5 is a soft label to be used to calculate the Siamese loss L_siamese as shown in Eq.8.

---

> ### Author Response · Authors · 2021-08-10
> **provide more diverse showcases**
>
> In Figure 4, the contextualized knowledge between internal clips in the same video is well provided. As we can see, it is unavoidable to lose some visual contents when sparsely sampling from the video. However, we can obtain the absent content from the siamese clips due to different sampling indices and duration, like the bird appearance in the siamese clips. We call such complementary information of internal clips as interdependent relationship. We will provide more showcases in our final version to better understand the benefits of learning from the inside.

---

> ### Author Response · Authors · 2021-08-10
> **More ablation studies**
>
> For number frames in anchor clips: We use the anchor clip with 5-16 frames according to the video length based on different datasets, such as 5 frames for some short-range videos (e.g., TGIF-QA) and 16 frames for long-range videos (e.g., ActivityNet-QA). We also set the anchor clip with 5, 10, 16 frames to analyze the effect of the number of frames in anchor clips. We conduct this ablation study on the MSVD-QA dataset. The anchor clip with 5 frames for our proposed method gets 44.9%. When we increase the frames to 10 and 16, it achieves 45.5% and 45.7, respectively. As we can see, increasing the frames can really bring gains. But the saturation of the improvement is also apparent. To balance the computation cost and the improvement, we set the frame number of the anchor clip on MSVD-QA as 10.
>
> For balancing term of integrating weights: We have conducted experiments on the MSVD-QA dataset to analyze the effect for these learnable weights (W1 and W2). Our SiaSamRea without W1 performs 44.9% on MSVD-QA, which heavily drops 0.6% performance compared with our whole SiaSamRea (45.5%). One possible reason is that the W1 can make the propagated prediction matrix learnable, and adaptively propagate the predictions between all the samples. When performing our model without W2, it achieves 45.2%. The performance without W2 is slightly decreasing (~0.3%). When it comes to our method without using W1 and W2, it gets 44.5%. As we can see, the W1 and W2 both can bright improvement. Hence, it can further demonstrate the importance and necessity for the generation of expected soft labels by correctly merging several predicted candidates.

---

> ### Author Response · Authors · 2021-08-10
> **ClipBERT on TGIF-QA in Table 5**
>
> There is a total of 71739 videos in the TGIF-QA dataset. Because our proposed method aims to model the long-range contextualized knowledge via Siamese samples. The videos with less than 40 frames do not meet our requirements in the training. We filter out less than 40 frames of video during the pre-training to remain long-range temporal videos. In the TGIF-QA dataset, the percentage of videos with less than 40 frames is ~56% (40333/71739). Hence, we use the remaining 31406 TGIF videos for our pre-training, which is less than the ClipBERT. The ClipBERT uses a total of 71739 TGIF-QA videos to finetune the four tasks, which is different from our pre-training on the TGIF-QA dataset.
>
> To fairly compare the effect of ClipBERT with our proposed method, we conduct extra experiments to finetune the ClipBERT on the filtered data and test it on the four tasks. We mark the ClipBERT finetuned on our filtered data as ClipBERT*. The ClipBERT* gets 75.1% on Action, 80.8% on Trans., 56.3% on Frame and 4.07 on Count. As we can see, the ClipBERT* can achieve comparable results compared with other SOTA methods. It can prove the effectiveness of the pre-trained weights from COCO Captions and Visual Genome Captions. However, our proposed method still outperforms the ClipBERT* on four tasks based on the same filtered data (79.7% on Action, 85.3% on Trans., 60.2% on Frame and 3.61 on Count). It can further demonstrate the superior ability of our idea by using the Siamese clips during the training. Our proposed method achieves better performance with less data. It also validates the huge potential of inside data once there is a suitable way to explore them.
>
> We also validate the ClipBERT on other benchmark datasets by using our experimental setting. It achieves 39.5% on MSVD-QA, 38.0% on ActivityNet-QA and 82.5% on How2QA. As we can see, ClipBERT performs comparable results on the VideoQA tasks thanks to their pre-trained knowledge. Our proposed method improves the model by mining the potential information of internal data, which is demonstrated effectively on five datasets.

---

### Official Review · Reviewer_a2XV · 2021-07-20

**Rating:** 5
**Confidence:** 4

**Summary:**

This paper proposes a novel approach, Siamese Sampling and Reasoning (SiaSamRea), to give self-taught guidance information as soft-labels to train the models via sampling several similar short-clips within the same video for the task of video QA. The authors provide experimental results to validate the effectiveness with new state-of-the-art performances on five benchmark datasets: MSRVTT-QA, MSVD-QA, How2QA, ActivityNet-QA and TGIF-QA.

**Ethical Concerns:**

Basically, Visual Question Answering systems including video QA provide answers based on arbitrary questions, especially open-domain QA.
If the questions include certain ethical values, how does this system generate answers?


**Limitations And Societal Impact:**

There is no limitations and societal impact in the manuscript.
How about mentioning the potential risks of the proposed video QA systems being trained with social biases on image data from the public or the industry?


**Main Review:**

The task of video QA is one of important and interesting problems in visual-linguistic AI. Since computational cost can be large for video understanding to handle arbitrary questions related to the video contents, it is inevitable to take certain sampling strategies. For example, one of recent works, ClipBERT, provides good approaches with sparse clip sampling. On the other hand, the proposed approach utilizes more sample clips that are similar but not same. With the aggregated features of the samples, the authors propose the methods to improve the models without supervision. The inductive bias of this work seems that “clips with high visual similarities are expected to have more consistent predictions on their class probabilities.” It can be achieved via Siamese sampling, sampling more short-clips near the anchor clip. From the assumption, the experimental results seem to show new state-of-the-art performances on some benchmark datatsets such as MSRVTT-QA, MSVD-QA, How2QA and ActivityNet-QA.

However, this paper has some weak points.
Firstly, the proposed methods seem to be based on ClipBERT [22] mostly. Following the manuscript, most parts of the architectures come from ClipBERT including video encoder, text encoder, and multi-modal transformer as a model of initial weights. I think the result on ClipBERT should be reported on all of the tested benchmark datasets. But, the results on MSRVTT-QA only are reported in Table 3. In this case, it is not clearly validated that the proposed method gives positive impacts. Moreover, the result on ClipBERT on TGIF-QA was reported in [22], and they are better than the result of the proposed approach.
Secondly, it is not clear that all of experiments performed fairly on all of comparative methods. The models in this paper are pretrained with TGIF-QA dataset from ClipBERTs as initial models, and the fine-tuning is done for the downstream tasks. On the other hands, it seems that the other methods were trained with their datasets and the strategies. If it is not easy to be performed properly, it would be better to clarify their experimental settings briefly in the manuscript.
Thirdly, it would be more persuasive to survey self-supervised learning methods in related work section.

Personally, I think the term Siamese is not good choice on this concept. It might lead to confusion.
As a suggestion, it would be better to clearly formulate and explain the proposed methods if finding and utilizing analogous machine learning frameworks, e.g., multi-view learning [1*,2*]. Probably, it will help the authors figure out pros and cons, the potentiality and the limitation, and find better directions to improve.



* Minor
   -	line 139, spare -> sparse


[1*] H. Su et al., Multi-view Convolutional Neural Networks for 3D Shape Recognition, ICCV 2015.

[2*] B. Xiong et al., Multiview Pseudo-Labeling for Semi-supervised Learning from Video, Technical Report, https://arxiv.org/abs/2104.00682.

----------------------------
### Post Rebuttal

I have read all of other reviews and the authors' response.
The rebuttal resolves most of my concerns.
However, I think that there is still remained necessity to improve the manuscript in presentation, discussion and reproducibility.
I keep my score as it was.

**Time Spent Reviewing:**

5

---

> ### Author Response · Authors · 2021-08-10
> **Limitations and potential risks**
>
> One of the limitations of our method is the generalization for different tasks. Currently, we mainly focus on the VideoQA task. We think there are more video-related tasks will be considered in our further work.
>
> One of the potential risks of the proposed system is the data biases. Our system tends to distinguish the wedding scene with a white wedding dress. As we know, the white color is a classical topic for western weddings. However, at the eastern wedding, the popular color should be red. And eastern people in the eastern funeral usually wear white clothes. Hence, our system sometimes mistakenly judges the eastern funeral as a wedding.

---

> ### Author Response · Authors · 2021-08-10
> **Add related work and modify siamese to better understanding**
>
> We will add more surveys of self-supervised learning methods in the related work section. We will carefully consider the name definition (e.g., Siamese) to make the formulation more distinct. Thanks for your suggestion.

---

> ### Author Response · Authors · 2021-08-10
> **Comparison with ClipBERT**
>
> There is a total of 71739 videos in the TGIF-QA dataset. Because our proposed method aims to model the long-range contextualized knowledge via Siamese samples. The videos with less than 40 frames do not meet our requirements in the training. We filter out less than 40 frames of video during the pre-training to remain long-range temporal videos. In the TGIF-QA dataset, the percentage of videos with less than 40 frames is ~56% (40333/71739). Hence, we use the remaining 31406 TGIF videos for our pre-training, which is less than the ClipBERT. The ClipBERT uses a total of 71739 TGIF-QA videos to finetune the four tasks, which is different from our pre-training on the TGIF-QA dataset.
>
> To fairly compare the effect of ClipBERT with our proposed method, we conduct extra experiments to finetune the ClipBERT on the filtered data and test it on the four tasks. We mark the ClipBERT finetuned on our filtered data as ClipBERT*. The ClipBERT* gets 75.1% on Action, 80.8% on Trans., 56.3% on Frame and 4.07 on Count. As we can see, the ClipBERT* can achieve comparable results compared with other SOTA methods. It can prove the effectiveness of the pre-trained weights from COCO Captions and Visual Genome Captions. However, our proposed method still outperforms the ClipBERT* on four tasks based on the same filtered data (79.7% on Action, 85.3% on Trans., 60.2% on Frame and 3.61 on Count). It can further demonstrate the superior ability of our idea by using the Siamese clips during the training. Our proposed method achieves better performance with less data. It also validates the huge potential of inside data once there is a suitable way to explore them.
>
> We also validate the ClipBERT on other benchmark datasets by using our experimental setting. It achieves 39.5% on MSVD-QA, 38.0% on ActivityNet-QA and 82.5% on How2QA. As we can see, ClipBERT performs comparable results on the VideoQA tasks thanks to their pre-trained knowledge. Our proposed method improves the model by mining the potential information of internal data, which is demonstrated effectively on five datasets.

---

### Author Response · Authors · 2021-08-20
**Thanks for all your comments and look forward to post-rebuttal feedbacks!**

Dear AC and all reviewers:

Thanks again for all of your constructive suggestions, which have helped us improved the quality and clarity of the paper!

Since the discussion phase has started for over one week, we have not heard any post-rebuttal response yet.

Please don’t hesitate to let us know if there are any additional clarifications or experiments that we can offer, as we would love to convince you of the merits of the paper. We appreciate your suggestions.

Best regards,

Authors of NeurIPS 2021 Submission #9197

---

### Decision · Program_Chairs · 2021-09-28

**Decision:**

Accept (Poster)

**Comment:**

After the authors’ rebuttal, intensive discussion followed between reviewers, and finally the reviewers’ consensus has reached to the reject opinion for this submission at this time. All agree that the main idea of this work is interesting, and they do not want to be strongly against this work. However, the reviewers also acknowledged that there is much room for improvement to be published. For example, the paper needs more clarifications and details about where the improvements come from, ClipBERT should be compared in different scenarios, and better benchmark datasets may be necessary.

**Consistency Experiment:**

NeurIPS has a long history of experimentation. In 2014, NeurIPS ran an experiment in which 10% of submissions were reviewed by two independent committees to quantify the randomness in the review process. This year, we repeated a variant of this experiment to see how the quality of the review process has changed over time.  This paper was part of the experiment and was therefore assigned to two committees (consisting of reviewers, an Area Chair, and a Senior Area Chair) that reached independent decisions.  If both committees made the same recommendation, this recommendation was followed. If a single committee recommended acceptance, the paper was accepted (with the exception of a few cases in which the other committee identified what we considered a fatal flaw, e.g., an error in a key result).

This copy’s committee reached the following decision: **Reject**

The other committee assigned to the paper recommended **Accept (Poster)**.  You can find the other set of reviews, along with any follow up discussion with the authors here:
https://openreview.net/forum?id=lDVeaQIScg